# Interpreting Adversarial Robustness: A View from Decision Surface in Input Space

## Abstract

One popular hypothesis of neural network generalization is that the flat local minima of loss surface in parameter space leads to good generalization. However, we demonstrate that loss surface in parameter space has no obvious relationship with generalization, especially under adversarial settings. Through visualizing decision surfaces in both parameter space and input space, we instead show that the geometry property of decision surface in input space correlates well with the adversarial robustness. We then propose an adversarial robustness indicator, which can evaluate a neural network's intrinsic robustness property without testing its accuracy under adversarial attacks. Guided by it, we further propose our robust training method. Without involving adversarial training, our method could enhance network's intrinsic adversarial robustness against various adversarial attacks.

## 1 Introduction

It is commonly believed that a neural network's generalization is correlated to its geometric properties of the loss surface, i.e. the flatness of the local minima in parameter space (Kawaguchi (2016); Im et al. (2017)). For example, Keskar et al. (2017) showed that a sharp local minima brings poorer neural network generalization. Li et al. (2017) visualized and demonstrated that the outstanding performance of ResNet comes from its "wide valley" of the local minima on the loss surface. Chaudhari et al. (2017) then proposed Entropy-SGD, which utilizes the "wide valleys" property to guide the neural network training for better generalization.

However, such a generalization estimation approach is challenged by adversarial examples recently (Szegedy et al. (2013)): even models with good generalization may still suffer from adversarial examples, resulting in extremely low accuracy. For example, ResNet model usually converges to a wide and flat local minima on loss surface in parameter space as visualized in Fig. 1(a), which indicates good test accuracy according to the mentioned evaluation approach. When testing adversarial examples, the model's accuracy is significantly defected by the adversarial noises, which are small, but can cause dramatically loss increments. Therefore, the conventional generalization estimation in parameter space fails under adversarial settings, and how to estimate the generalization over adversarial examples, i.e. the adversarial robustness, remains a significant challenge.

Fortunately, the loss increments introduced by adversarial noises can be well reflected in the loss surface in input space as visualized in Fig. 1(b), where the non-smoothness indicates high sensitivity

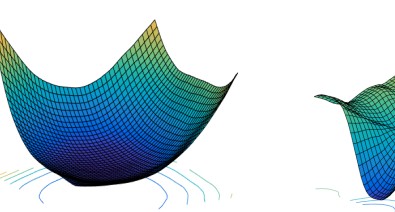
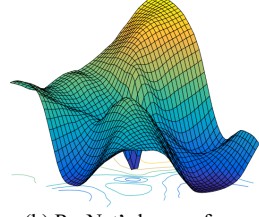

(a) ResNet's loss surface in parameter space*

(b) ResNet's loss surface in input space*

\* The magnitude of the two surfaces indicates the loss value, while the left surface changes with neural network parameters (e.g. weights and biases update), and the right surface changes with input variation (e.g. noises on test images).

Figure 1: (a) ResNet's loss surface on local minima in parameter space has wide and flat geometry, indicating optimal accuracy. (b) While in input space, the loss surface demonstrates significant non-smooth variation.

to the adversarial noises. The differences in Fig. 1 suggest the ineffectiveness of generalization estimation in parameter space and the potential in input space. Therefore, different from prior works focusing on parameter space, we will explore the robustness estimation mainly in input space.

In this work, we have the following contributions:

- We analyze the geometric properties of the loss surface in both parameter and input space. We demonstrate that the input space is more essential in evaluating the generalization and adversarial robustness of a neural network;

- We reveal the shared mechanisms of various adversarial attack methods. To do so, we first extend the concept of loss surface to decision surface for clearer geometry visualization. By visualizing the adversarial attack trajectory on decision surfaces in input space, we reveal that various adversarial attacks are all utilizing the decision surface geometry properties to cross the decision boundary within least distance.

- We then formalize the adversarial robustness indicator by involving the geometric properties of Jacobian and Hessian's eigenvalues. Such an indicator can effectively evaluate a neural network's intrinsic robustness property against various adversarial attacks without field accuracy testing of massive adversarial examples; It also concludes that the wide and flat plateau of decision surface in input space enables better generalization and robustness.

- We also propose a robust training method guided by our adversarial robustness indicator, which aims to smooth the decision surfaces and enhances adversarial robustness by regulating the Jacobian in training process.

Our robustness estimation approach has provable relationship with neural network's robustness performance and geometry properties in input space. This enables us to evaluate adversarial robustness of a neural network, without conducting massive adversarial attacks for field test. Guided by such an estimation approach, our robust training method could also effectively enhance the neural network against various adversarial attacks without involving the time-consuming adversarial training.

## 2  INEFFECTIVENESS OF ADVERSARIAL ROBUSTNESS ESTIMATION FROM THE LOSS SURFACE IN PARAMETER SPACE

In this section, we compare the effectiveness of the adversarial robustness estimation from the loss surface in both parameter and input space with an effective visualization method.

### 2.1  VISUALIZATION NEURAL NETWORK LOSS SURFACE BY PROJECTION

Given a neural network with a loss function $F(\theta, x)$, where $\theta$ is neural network parameters (weight and bias) and $x$ is the input. As the function inputs are usually in high-dimensional space, direct visualization analysis on the loss surface is impossible. Therefore, following the methods proposed by Goodfellow et al. (2015) and Li et al. (2017), we project the high-dimensional loss surface into a low-dimensional space, e.g. 2D hyperplane to visualize it. In such methods, two projection vectors $\alpha$ and $\beta$ are chosen and normalized as the base vectors for $x$ and $y$ axes. Then given an starting point $o$, the points around it are interpolated and corresponding loss values can be calculated:

$$V(i, j, \alpha, \beta) = F(o + i \cdot \alpha + j \cdot \beta) \tag{1}$$

Here, the original point $o$ in function $F$ could be either in parameter space, which is mostly studied by prior work (e.g. Li et al. (2017); Im et al. (2017)), or in input space, which is our major focus in this paper. The coordinate $(i, j)$ denotes how far the original point moves along $\alpha$ and $\beta$ direction. After calculating enough points' loss values, the function $F$ with high-dimensional inputs could be projected to the chosen hyperplane formed by vector $\alpha$ and $\beta$.

Fig. 1 has already shown two visualized loss surface examples in both parameter space and input space, which give an intuition of the dramatic difference between the two approaches. In the next section, we will further demonstrate that the loss surface geometry in input space is more essential regarding the neural network robustness properties.

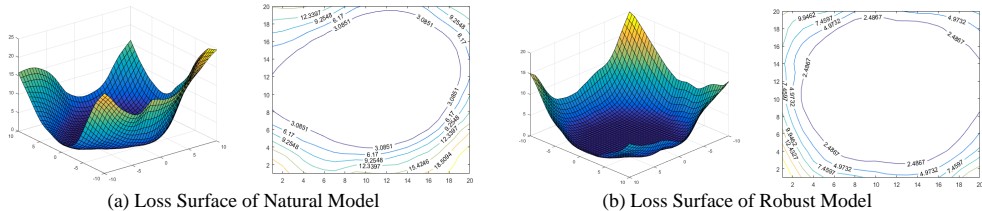

(a) Loss Surface of Natural Model          (b) Loss Surface of Robust Model

Figure 2: Loss Surfaces in parameter space (similar settings with Li et al. (2017)). There is no clear relation between the neural network robustness and loss surface geometry in parameter space.

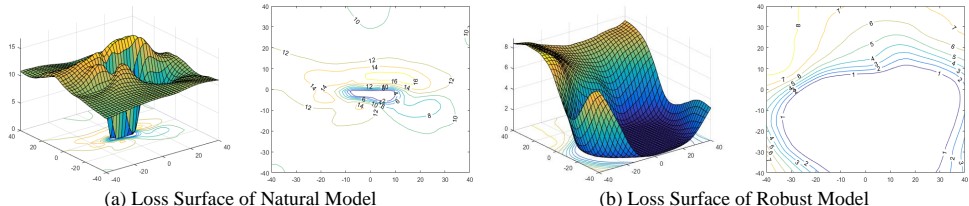

(a) Loss Surface of Natural Model          (b) Loss Surface of Robust Model

Figure 3: Loss Surface in input space (our main focus). There are distinct difference between two models with different degrees of robustness. Obviously, network robustness is highly correlated to the loss surface geometry in input space, instead of parameter space.

## 2.2 THE LOSS SURFACE IN PARAMETER SURFACE VS. INPUT SPACE

To prove our statement, we examine the robustness of a pair of neural networks with the same ResNet model setting, but trained with natural process and Min-Max robust training scheme respectively. Both neural networks could achieve optimal accuracy ($\sim 90\%$) on CIFAR10 dataset. However, their adversarial robustness degrees are significantly different: 0.0% and 44.71% accuracy under $\ell_\infty = 8$ adversarial attacks. To analyze such a difference, the loss surfaces and corresponding contour maps are visualized in both parameter space (as shown in Fig. 2) and input space (as shown in Fig. 3). Here the $z$-axis of 3D visualization denotes the loss values (as well as the numbers on contour lines in 2D visualization), and the $x$ and $y$ axes are the corresponding projection vectors $\alpha$ and $\beta$.

When illustrated in parameter space, both neural networks' loss surfaces on local minima are wide and flat, which align well with their high accuracy as stated in (Li et al. (2017)). But comparing Fig. 2 (a) and (b), even though the two neural networks have distinct degrees of robustness, there is no obvious difference between their loss surfaces in parameter space. More examples could be found in Appendix 8.5, which also indicates the limitations of parameter space loss to describe robustness.

However when illustrated in input space as shown in Fig. 3, obvious differences emerge between the natural and robust neural networks: (1) Based on the 3D surface visualization, the natural neural network's loss surface in input space has a deep and sharp bottom, while the local minima one the loss surface of the robust neural network is much flatter; (2) Based on the contour map visualization, we show that the original inputs in the natural neural network's surface locate in a very small valley, while the robust one shows a wide area. Thus, in the natural neural network's case, once some small perturbations are injected into the inputs and move the inputs out of the small valley, the function loss will significantly increase and the prediction result could be easily flipped.

By comparison of parameter space and input space loss surfaces, we demonstrate the potential advantages of using loss surfaces in input space than in parameter space. Therefore, we clarify that in terms of generalization and adversarial robustness, we should not only focus on the so-called "wide valley" of loss surfaces in parameter space, but also in input space. In the next section, we will visualize the adversarial attack trajectory and further demonstrate the close relation between adversarial vulnerability and decision surface geometry in input space.

## 3 REVEALING ADVERSARIAL ATTACKS' MECHANISM THROUGH DECISION SURFACE ANALYSIS

Previously, we showed the potential of adversarial robustness estimation in input space. In this section, we further explore the adversarial attacks' mechanism with input space decision surface.

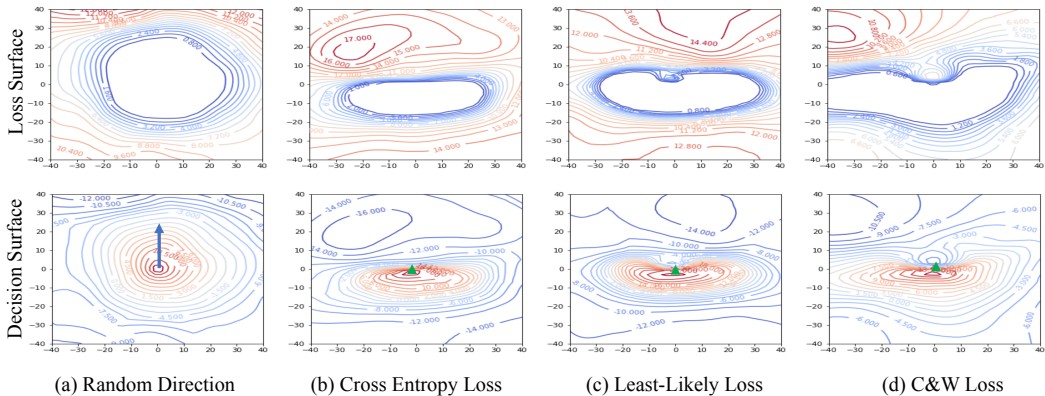

|                |                |                |             |
|----------------|----------------|----------------|-------------|
| (a) Random Direction | (b) Cross Entropy Loss | (c) Least-Likely Loss | (d) C&W Loss |

Figure 4: Comparison of cross entropy based loss surface (first row) and our decision surface (second row). In (a), the projection vectors are randomly selected. In (b)-(d), $y$-axis projection vector is the descent direction of corresponding objective function of various adversarial attacks. Along both axes we use step size = 1 on the pixel range ($0 \sim 255$). Clearly, all three adversarial attacks are utilizing the geometry information of decision surface to find the shortest paths (indicated by arrows and triangles) to cross the decision boundaries.

## 3.1 EXTEND LOSS SURFACE TO DECISION SURFACE

Directly visualizing loss surfaces in input space has certain disadvantages: The major issue is that there is no explicit decision boundary for the correct or wrong prediction for a given input image; Another deficiency of cross entropy based loss surface is that it cannot well demonstrate the geometry, especially when the loss is relatively low [1]. This usually causes large blank regions with no useful information in visualization. To resolve these problems, we extend the concept of loss surface to decision surface, which enriches the geometry information and offers clear decision boundary.

Here, we introduce the definition of neural network decision boundary and decision surface. For one input image $x$ with label $y_t$, the decision boundary of a neural network is defined as following:

$$L(x) = Z(x)_t \; - \; max\{Z(x)_i, \; i \neq t\} = 0. \tag{2}$$

$Z(x)$ is the logit layer output before softmax layer, and $t$ is the true class of input $x$. The decision function $L(x)$ evaluates the confidence of prediction, i.e. how much correct logit output exceeds the max incorrect logit. In correct prediction cases, $L(x)$ should always be positive and higher value is often better (different from cross-entropy loss that lower is better). $L(x) = 0$ indicates the equal confidence in correct and wrong class, and thus is the decision boundary between correct and wrong prediction. The surface formed by function $L(x)$ is called the decision surface to distinguish from cross entropy based loss surface, also because it contains the explicit decision boundary.

Fig. 4 compares the cross entropy based loss surface visualization (first row) and the decision surface visualization (second row). The decision surfaces can well resemble the loss surfaces in their informative areas but also include the confidence information in the blank area of the loss surfaces. Meanwhile, the explicit decision boundary, i.e. contour line $L(x) = 0$, enables us to clearly see when network decision changes, which is very useful in analyzing adversarial examples as we will show next. In the following paper, we will use decision surface visualization as default settings.

## 3.2 SHARED MECHANISM OF VARIOUS ADVERSARIAL ATTACKS

As shown in Eq. 1, we could project the decision surface to a hyperplane composed of two base vectors – $\alpha$ and $\beta$. Therefore, using adversarial attacking direction $\delta$ as the projection vector $\beta$ ($y$-axis), we could visualize the adversarially projected decision surfaces and the corresponding attack trajectory along the $y$-axis direction.

For generality, we compare four cases with different projection vector $\beta$: The first one is random direction, but the other three are produced from three representative adversarial attack objective functions: cross-entropy non-targeted loss, cross-entropy targeted loss (least-likely class) (Kurakin

---

[1]For the same input image, neural network prediction with high confidence can have similar loss with low confidence prediction due to the non-linear operations. Detailed analysis could be find in the Appendix.

et al. (2016)), and C&W loss (Carlini & Wagner (2017)):

$$\beta = sign(\nabla_x \, loss(x)), \;\; where \; loss(x) \in$$
$$\{ \, y_t log(softmax(Z)), \; y_l log(softmax(Z)) \, , \; max\{Z(x)_i, i \neq t\} - Z(x)_t. \, \}, \tag{3}$$

where $y_t$ is the true class label, $y_l$ is least likely class label (both one-hot).

The decision surface and adversarial trajectory visualization results are shown in Fig. 4. The length of blue and green arrows denote the distance needed to cross the decision boundary. In random projection (a), the length of green arrows is long. This indicates that towards a random direction, the original input is far from neural network's decision boundary or wrong classification regions with $L(x) < 0$. This explains the common sense that natural images with random noises won't degrade neural network accuracy significantly. However, in adversarially projected hyperplane (b-d), wrong regions are much closer indicated by extremely short arrows: Towards $y$-axis adversarial direction, adversarial examples could be easily found and even within $\ell_{\inf}(\delta) = 1$.

Comparing different adversarial attack trajectories in Fig. 4(b)-(d), we could find that they all demonstrate similar behaviors even though their objective functions are designed differently: All $y$-axis attack directions show extremely dense contour lines. This denotes the steepest descent direction to cross decision boundary $L = 0$ and to enter wrong classification regions. Therefore, we can conclude the shared mechanism by different adversarial attacks, which is to utilize the decision surface geometry information to cross the decision boundary within shortest distance.

Meanwhile, our visualization results reveal the nature of adversarial examples: Although a neural network's training loss in parameter space seems to converge well after model training, there still exist large regions of points that the neural network fails to classify correctly (proved by the large negative regions on the adversarial projected hyperplane). And some of these regions are extremely close to the original input points (some even within $\ell_{\inf}(\delta) = 1$ distance). Since the data points in such regions are in the close neighborhood of the natural input images, they seem no difference by human vision, which is conventionally recognized as adversarial examples.

Therefore, we conclude that rather than being "crafted" by adversarial attacks, adversarial examples are "naturally existed" points. Rather than defending "adversarial attacks", the essential solution of robustness enhancement is to solve the "neighborhood under-fitting" issue of neural networks. We then propose an adversarial robustness evaluation approach, which uses decision surface's differential geometry property to interpret and evaluate the neural network robustness.

## 4 ADVERSARIAL ROBUSTNESS INDICATOR WITH DECISION SURFACE GEOMETRY

### 4.1 THEORETICAL ROBUSTNESS BOUND BASED ON SECOND-ORDER TAYLOR EXPANSION

Suppose neural network decision function $L(\theta, x)$ is second-order differentiable. We have noticed that the neural network decision surface in input space captures more information about adversarial vulnerability. Since $L(\theta, x)$ has no explicit formulation, we could utilize the second-order Taylor Approximation *w.r.t* input $x$ to approximate it within $x's$ neighborhood:

$$L(\theta, x + \Delta x) = L(\theta, x) + J\Delta x + \frac{1}{2!} \, \Delta x^T H \Delta x, \tag{4}$$

where $\theta$ is the parameters of the neural network. The Jacobian vector $J$ is of the same dimension with $x$ , and $x$ is the input feature vector. And Hessian matrix $H$ is a square matrix of second-order partial derivatives of $F(\theta, x)$ with regard to $x$.

Given a correctly classified input $x$ with confidence $L(\theta, x) = t \; (t > 0)$. In adversarial settings, the adversarial robustness of neural network means that given a feasible set $S$, e.g. $\ell_\infty$ constraints, all perturbations in this set cannot change the decision. Formally, it can be defined as following:

$$sign(L(\theta, x + \delta)) = sign(L(\theta, x)), \; \forall \delta \in S. \tag{5}$$

To connect this objective function with our decision surface, we enforce a new constraint:

$$|L(\theta, x + \delta) - L(\theta, x)| < t, \; \forall \delta \in S. \tag{6}$$

This leads to $L(\theta, x + \delta) \in (0, 2t)$, thus $L(\theta, x + \delta)$ has the same sign with $L(\theta, x)$. Clearly, Eq. 5 is strictly guaranteed. Meanwhile, this formulation enforces stronger constraints: it means when neural network predicts, its neighborhood points should not only share the same decision but also have similar confidence bounded by absolute difference $t$, similar with *mixup* (Zhang et al. (2017)).

Then, combining Taylor Approximation in Eq. 4 and Eq. 6, the following inequality can be derived:

$$\max_{\delta \in S}( \, |J \cdot \delta + \frac{1}{2} \, \delta^T H \delta| \, ) < t. \tag{7}$$

Since Hessian matrix $H$ is orthogonally diagonalizable, it could be decomposed by eigen-decomposition: $H = E\Lambda E^T$. $E$ is the eigenvector matrix $[e_0, e_1, \cdots, e_n]$ composed of $H$'s eigenvectors $e_i$, and $\Lambda$ is the diagonal eigenvalue matrix with only $H$'s eigenvalues in the diagonal as the non-zero entries. Let $y = E^T \delta$, we have:

$$\delta^T H \delta = \delta^T E \Lambda E^T \delta = (E^T \delta)^T \Lambda (E^T \delta) = y^T \Lambda \, y. \tag{8}$$

Therefore, we could show that the upper bound of Eq. 7 is:

$$|J \cdot \delta + \frac{1}{2} \, \delta^T H \delta| \le |J \cdot \delta| + \frac{1}{2}|y^T \Lambda \, y| \le \sum_{i=1}^{n} |J_i| \cdot |\delta_i| + \frac{1}{2} \sum_{i=1}^{n} |\lambda_i| \cdot |y_i^2|. \tag{9}$$

Here $\delta_i$ and $y_i$ are the entries of vector $\delta$ and $y$, which depend on the choice of perturbation $\delta$. $J_i$ is the entry of Jacobian vector, and $\lambda_i$ is the eigenvalue of Hessian $H$. Intuitively, given the constraints on $\delta$ (e.g. $\ell_\infty$ constraints), the upper bound highly depends on the Jacobian and Hessian matrix. As long as the magnitude of every $J_i$ and eigenvalue $\lambda_i$ of Hessian $H$ could be controlled to the minimum, e.g. near zeros, the influence of perturbation $\delta$ can be constrained to a certain range, i.e. robust to any noises. Therefore, the average magnitude of these two parameter sets of a neural network could be defined as its robustness indicator.

## 4.2 The Geometric Explanation of Robustness Indicator

As shown in Eq. 9, model robustness highly relies on the magnitude of Jacobian entries and eigenvalues of Hessian. In differential geometry, these parameters has their specific geometry meaning: For a multi-variable function $F(x)$, Jacobian entry $J_i$ measures the slope of the tangent vector at point $x$ along $x_i$ axis, where low value denotes flat neighborhood. Therefore, it's easy to understand that smaller Jacobian leads to a flat minima. Meanwhile, magnitude of eigenvalues $\lambda_i$ of Hessian denotes the curvature (Alain et al. (2018); Cheeger & Ebin (2008)), which is defined as:

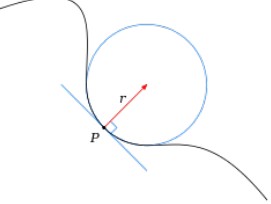

$$\kappa = 1/r. \tag{10}$$

$\kappa$ is the reciprocal of the radius $R$ of osculating circle at current point. The conception in simple 1-d case is shown in Fig. 5. Clearly, lower curvature (small eigenvalues) means that the hyperplane bends less, leading to a wider neighborhood of original point.

Figure 5: An 1-D illustration of slope and curvature

Based on the differential geometry meaning of Jacobian and Hessian, we could conclude that both constraints on Jacobian and Hessian in Eq. 9 appeal for a wider and flatter local neighborhood of input space decision surface (not parameter space). This is consistent with our preliminary visualization results in Fig. 3. Next, we will qualitatively and quantitatively demonstrate the effectiveness of our neural network robustness indicator.

## 4.3 Robustness Indicator Evaluation: A Case Study

**Decision Surface of Natural Model vs. Robust Model** In this section, we compare two pairs of robust and natural models on MNIST and CIFAR10. The two pair of models are released in MNIST/CIFAR adversarial challenges (Madry et al. (2018)) with same structure and comparable accuracy in natural test settings but different degree of robustness. The robust MNIST model is trained by Min-Max optimization and could achieve $\sim$88% accuracy under all attacks with the $\ell_\infty < 0.3$ constraints on a $(0, 1)$ pixel range, which is believed to be the currently most robust model

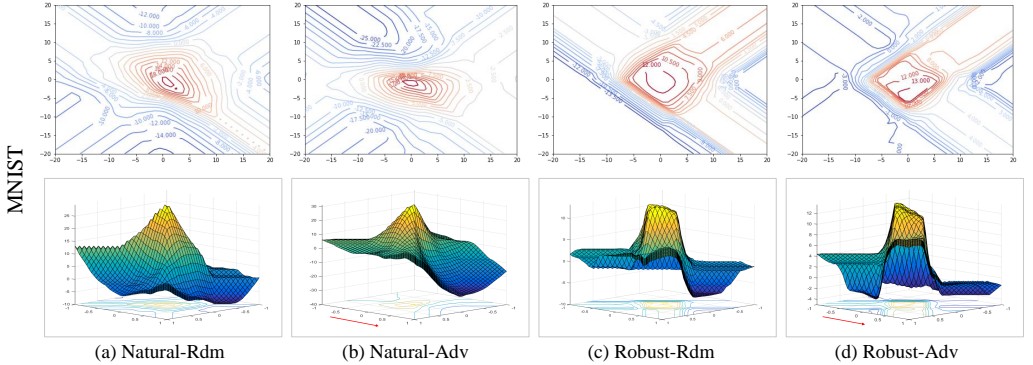

Figure 6: Decision Surface of the natural and robust model on MNIST. (a)-(d): natural model surface with random projection, adversarial projection; robust model surface with random projection, adversarial projection (red arrows denote the adversarial direction with step size = 0.05). On robust model surface, both original input and its neighbor points (within $\ell_\infty < 0.3$) locate on the flat plateau. However, natural surfaces usually show sharp peaks and thus are vulnerable to adversarial noises.

on MNIST dataset. By contrast, the natural model can be totally broken with 0.0% accuracy within same constraints. CIFAR models are same as Sec. 2. To prove our geometric robustness theory, we first visualize two pair of models' decision surfaces in Fig. 6 (MNIST), and Fig. 7 (CIFAR10).

From Fig. 6, we can find significant difference between natural and robust decision surface: On robust decision surfaces (c) and (d), whether we choose random or adversarial projection, all neighborhood points around the original input point locates on the high plateau with $L(x) > 0$. The surface in the neighborhood is rather flat with minimum slopes until it reaches $\ell_\infty < 0.3$ constraints, which is the given adversarial attack constraint. This explains its exceptional robustness against all adversarial attacks. By contrast, natural decision surfaces shows sharp peaks and large slopes, on which decision confidence could quickly drop to negative along the $y$-axis. For CIFAR10 models as shown in Fig. 7, similar conclusions could be drawn that the degree of adversarial robustness depends on how well models could fit the neighborhood of input points, and a flat and wide plateau around the original points on decision surface is one of the most desired properties of a robust model [2].

**Jacobian and Hessian Statistics Analysis** We also analyze the statistics of previous natural and robust MNIST model's Jacobian and Hessian matrix. We randomly take 100 input images from test set and calculate their Jacobian and Hessian using natural and robust model, respectively. The data distribution and visualization results are shown in Fig. 8.

First, the $\ell_1$ norm of robust model's Jacobian is much smaller than natural models: average $\ell_1$ norm of robust model's Jacobian is about $\sim 50$, ten times less than $\sim 500$ of natural model's. And $\ell_1$ norm of robust model's Hessian are $0.53$, two times less than $1.26$ of natural models'. Therefore, the statistics

---

[2]More examples could be found in Appendix.

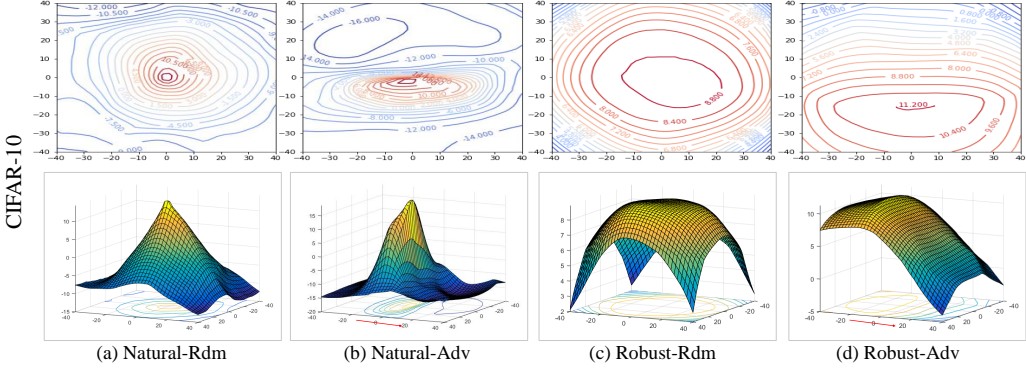

Figure 7: Comparison of Decision Surface of natural and robust model on CIFAR10 (step size = 1). As assumed, natural model's surface shows sharp peaks while robust model's shows flat plateau.

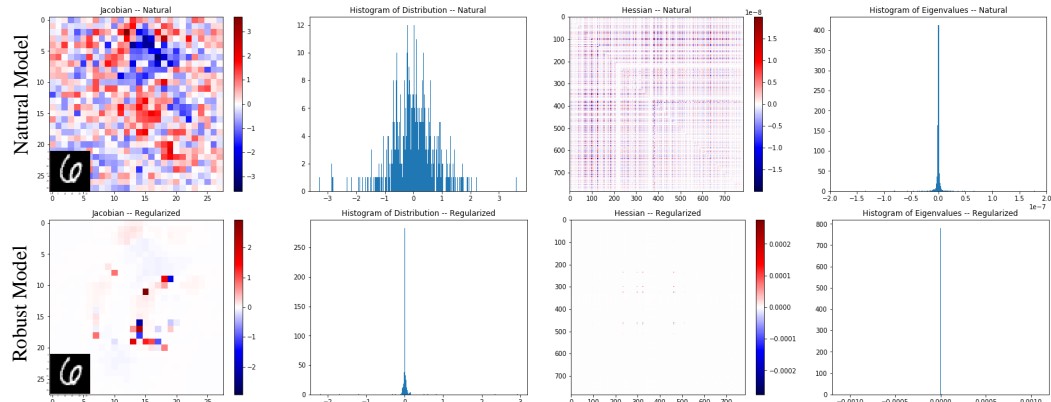

Figure 8: MINST models' Jacobian and Hessian visualization and analysis with randomly selected input image "6". Robust model's Jacobian and Hessian are more sparse, and have smaller $\ell_1$ norm.

of Jacobian and Hessian also verifies our robustness indicator theory in Eq. 9. Another significant difference is that compared to natural model, robust model's Jacobian and Hessian are more sparse. For natural and robust models, the ratio of zeros in Jacobian are 1.1% and 54.0% respectively. As for Hessian, the ratio of zeros in eigenvalues for natural and robust models are 65.6% and 97.1% respectively[3]. One intuitive explanation of the relationship between robustness with Jacobian sparsity is that natural model's Jacobian contains many un-meaningful but noisy gradients, while most of robust model's Jacobian non-zero entries concentrate on the main trace of the digits, or so-called main features. This assumption is based on our observation that for most input images, robust model's Jacobian could precisely capture the main trace of digits or main patterns of the object, as shown in Fig. 8. In such ways, if adversarial noise is uniformly injected into every pixel of the input image, only a small portion of them will be likely to influence decision, thus more robust to perturbations.

## 5    TOWARD ROBUSTNESS ENHANCEMENT AGAINST ADVERSARIAL ATTACKS

### 5.1    ROBUST TRAINING FOR SMOOTHING DECISION SURFACE IN INPUT SPACE

As shown in Eq. 9, better robustness of neural networks needs lower magnitude or zero Jacobian and Hessian. Here, we use Jacobian of our decision function $L(\theta, x)$ *w.r.t* $x$. Therefore to improve the network robustness and flatten the local minima, we propose a simple yet effective approach: add a regularizer on the Jacobian of decision loss $L(x)$ in network training process. This could be done through *double backpropagation* (Drucker & Le Cun (1991)). As for Hessian eigenvalue, calculating the Hessian eigenvalues takes $\Theta(k^3)$ complexity ($k$ is parameter space dimension). Currently there is no efficient way to regulate it in neural network context. Possible techniques to use are Hessian diagonal approximation (Martens (2010); Becker et al. (1988)), which we leave as future work.

To regulate the Jacobian of decision function $L(\theta, x)$, we could add a regularization term on $\partial L(\theta, x)/\partial x$ to network training loss $Loss_\theta(x)$:

$$Loss_\theta(x) = L_{ce} \; + \; c \; \cdot \; L_{norm}(\frac{\partial L(\theta, x)}{\partial x}), \tag{11}$$

where $L_{ce}$ is cross-entropy loss, and hyper parameter $c$ is the factor of penalty strength. For the regulation term $L_{norm}(\cdot)$, we can choose $\ell_1$, $\ell_2$ or $\ell_\infty$. As aforementioned, introducing the gradient loss into the training loss needs us to solve a second-order gradient computing problem. To solve this problem, double backpropagation (Drucker & Le Cun (1991)) is needed. The cross-entropy loss are first computed by forward-propagation, with the gradients then being calculated by backpropagation. Note here we need to calculate both $\partial L_{ce}/\partial\theta$ and $\partial L(\theta, x))/\partial x$ as required in Eq. 12. Then, to minimize the gradient loss, the second-order mixed partial derivative of gradient loss *w.r.t* $\theta$ is calculated. Note this mixed partial derivative is different from Hessian (which is pure second-order derivative of $x$ or $w$), and thus is calculable (Ororbia et al. (2016)). After this, a second

---

[3]Jacobian and Hessian matrix entries are mostly near zero but non-zero values, therefore we consider values below $10e$-3 (Jacobian) and $10e$-10 (Hessian) as zeros.

Table 1: Test Accuracy of adversarial examples on MNIST dataset (%)

| Models | Natural | FGSM | | | BIM | | | C&W | | |
|---|---|---|---|---|---|---|---|---|---|---|
| | | 0.1 | 0.2 | 0.3 | 0.1 | 0.2 | 0.3 | 0.1 | 0.2 | 0.3 |
| Natural Model | 99.1 | 67.3 | 12.9 | 4.7 | 22.5 | 0.0 | 0.0 | 21.6 | 0.0 | 0.0 |
| AdvTrain | 99.1 | 73.0 | 52.7 | 10.9 | 62.0 | 6.5 | 0.0 | 71.09 | 17.0 | 2.1 |
| CrossEntropy | 99.2 | 91.6 | 60.4 | 18.3 | 87.9 | 19.9 | 0.0 | 88.09 | 20.0 | 0.0 |
| Ours | 98.4 | 91.6 | 70.3 | 41.6 | 88.1 | 64.9 | 26.7 | 89.2 | 72.6 | 37.6 |

Table 2: Test Accuracy of adversarial examples on CIFAR10 dataset (%)

| Models | Natural | FGSM | | | BIM | | | C&W | | |
|---|---|---|---|---|---|---|---|---|---|---|
| | | 3 | 6 | 9 | 3 | 6 | 9 | 3 | 6 | 9 |
| Natural Model | 87.2 | 5.8 | 2.4 | 1.6 | 0.7 | 0.0 | 0.0 | 0.6 | 0.0 | 0.0 |
| AdvTrain* | 84.5 | 10.2 | 5.8 | 2.6 | 1.4 | 0.0 | 0.0 | 0.0 | 0.0 | 0.0 |
| CrossEntropy* | 86.2 | 19.1 | 9.5 | 6.1 | 2.6 | 0.7 | 0.4 | 2.1 | 1.5 | 1.4 |
| Ours | 84.2 | 59.8 | 41.9 | 31.0 | 54.6 | 29.5 | 20.3 | 53.7 | 29.8 | 20.1 |
| Ours+AdvTrain | 83.1 | 68.5 | 48.5 | 38.2 | 62.7 | 39.3 | 30.3 | 60.5 | 39.0 | 30.3 |
| MinMax* | 79.4 | 65.8 | 55.6 | 47.4 | 64.2 | 49.3 | 41.1 | 62.9 | 48.5 | 40.7 |

[*] AdvTraining denotes adversarial training (Kurakin et al. (2016)). CrossEntropy denotes the cross entropy gradient regularization (Ross & Doshi-Velez (2018)). MinMax denotes (Madry et al. (2018))'s method.

backpropagation operation is performed, and the weights of neural networks are updated according to gradient descent algorithm:

$$\theta' = \theta - lr \cdot \left(\frac{\partial L_{ce}}{\partial \theta}\right) - lr \cdot c\left(\frac{\partial L_{norm}(\partial L(\theta, x)/\partial x)}{\partial \theta}\right). \tag{12}$$

Compared to adversarial training based defense techniques, our proposed robust training method doesn't rely on adversarial example generation method, and thus is capable to defend against different adversarial attacks. The main extra computation overhead is the doubled backpropagation computation time, which we will show in the following experiments.

## 5.2 ROBUST TRAINING PERFORMANCE EVALUATION

For comparison of the effect of robustness enhancement, we conduct different previous techniques: adversarial training (Kurakin et al. (2016)), cross-entropy gradient regularization (Ross & Doshi-Velez (2018)), Min-Max training (Madry et al. (2018)), our method, and etc. Evaluated adversarial attacks include Fast Gradient Sign Method (FGSM), Basic Iterative Method (BIM) and C&W attack. Detailed experiment settings could be found in Appendix. Final results are shown in Table. 1 and 2.

On MNIST dataset, our proposed gradient regularization method can achieve $\sim 90\%$ accuracy under all considered attacks within $\ell_\infty = 0.1$ constraints. Cross entropy gradient regularization (Ross & Doshi-Velez (2018)) achieves similar robustness as ours within $\ell_\infty = 0.1$, but their robustness performance drops very fast when adversarial attacking strengths increases, e.g. under $\ell_\infty = 0.2, 0.3$ attacks. The reason is that the softmax and cross entropy operation introduces unnecessary non-linearity, in which case the gradients from cross entropy loss is already very small.

Improving robustness on CIFAR10 dataset is much harder than MNIST. State-of-the-art MinMax training achieves $\sim 40\%$ accuracy under strongest attacks in considered settings. But this method highly relies on huge amounts of adversarial data augmentation methods, which takes over 10 times of overhead during training process. By contrast, our method doesn't need adversarial example generation and can achieves comparable robustness under $\ell_\infty = 3$ constraints. We test the average time of each epoch for both natural training and gradient regularized training. Our time consumption is average 2.1 times than natural training per epoch. Notice that the robustness enhancement of our method becomes lower when $\ell_\infty$ becomes larger. This shows one limitation of gradient regularization methods: Our gradient regularization approach is based on Taylor approximation in a small neighborhood. When the adversarial examples exceeds the reasonable approximating range, the gradient regularization effect also exhausts. Empirically, we found robust training usually takes more epochs to converge: On CIFAR10, natural training takes about 30 epochs, our method usually need 100 epochs, and MinMax robust training takes over 400 epochs to converge in our implementation.

## 5.3 ANALYZING THE INPUT SPACE DECISION SURFACE AND STATISTICS

To test if our robust training method flattens the loca minima of decision surfaces, we also visualize and comapre natural and our model's decision surface, as shwon in Fig. 9. Compared to natural model's surface, our model clearly has wider local neighborhood and lower slopes as expected. Meanwhile, the statistics of Jacobian and Hessian on MNIST models also align well with our previous robustness indication: The average $\ell_1$ norm of Jacobian and Hessian of our models are 10 and 3 times less than natural model, respectively.

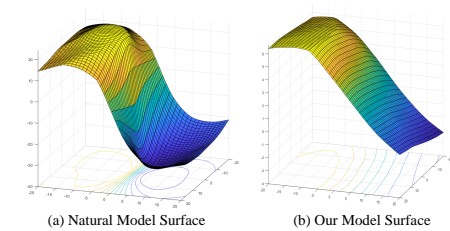

(a) Natural Model Surface          (b) Our Model Surface

Figure 9: Natural and our robust model surface.

Fig. 10 shows several examples and their Jacobian visualization of both natural and robust models (normalized to $0 \sim 255$ range for visualization). The robust model's Jacobian demonstrates better capability to capture the main feature of images on both MNIST and CIFAR10, as mentioned before.

## 6 RELATED WORK

Previous one popular hypothesis is that neural network's good generalization comes from flat local minima of the loss function in parameter space (Im et al. (2017); Keskar et al. (2017); Dinh et al. (2017); Kawaguchi (2016)). For example, Li et al. (2017) propose a visualize technique which establishes good connections between the minima geometry and generalization on ResNet. However, recently adversarial examples were introduced, which challenges the above generalization theory. Many adversarial attack methods are proposed (Szegedy et al. (2013); Kurakin et al. (2016); Carlini & Wagner (2017); Papernot et al. (2016a)). As for defense techniques, current defense techniques include adversarial training (Ian J. Goodfellow (2014)), defensive distillation (Papernot et al. (2016b)), parseval network (Cisse et al. (2017)), Min-Max robustness optimization (Madry et al. (2018)), adversarial logit pairing (Kannan et al. (2018)), and etc. Original adversarial training techniques augment the natural training samples with corresponding adversarial examples together with correct labels. Recently proposed Min-Max robustness optimization (Madry et al. (2018)) augments the training dataset with large amount of adversarial examples which cause the maximum loss increments within a $\ell_\infty$-norm ball, which is currently the strongest defense.

## 7 CONCLUSION

In this work, through visualizing network loss surface in parameter and input space, we point out the ineffectiveness of previous generalization theory under adversarial settings. Meanwhile, we show that adversarial examples are essentially the neighborhood under-fitting issue of neural networks in input space. We then derive the connection between network robustness and decision surface geometry as an indicator of the neural network's adversarial robustness. Guided by the indicator, we propose a practical robust training method, which involves no adversarial example generation. Extensive visualization results and experiments verify our theory and demonstrate the effectiveness of our proposed robustness enhancement method.

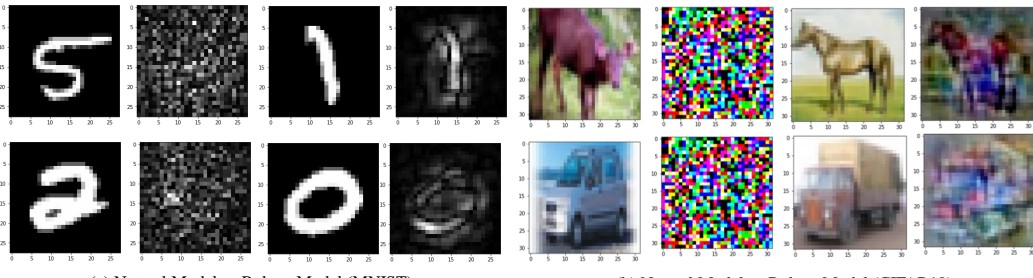

(a) Natural Model vs Robust Model (MNIST)          (b) Natural Model vs Robust Model (CIFAR10)

Figure 10: Jacobian Visualization on MNIST and CIFAR10. Through gradient regulation, neural network is more capable to capture the main pattern of input images.

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

# 8 APPENDIX

## 8.1 THE EXPLANATION OF INEFFECTIVENESS OF LOSS SURFACE WITH BLANKS

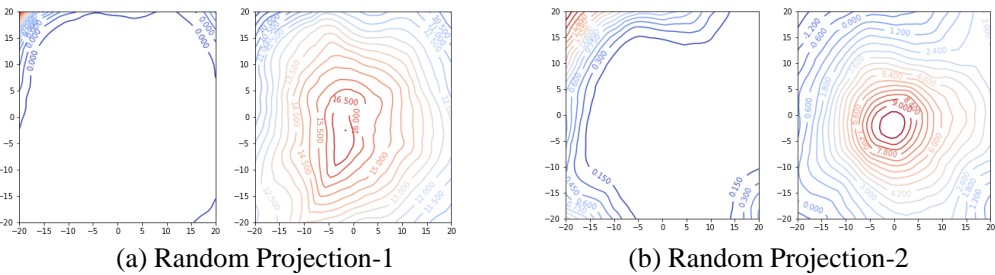

         (a) Random Projection-1            (b) Random Projection-2

Figure 11: Two pairs of random projections of loss surfaces and decision surfaces on the same images. We could clearly see that proposed decision surfaces can better demonstrate the geometry information compared to loss surfaces with usually large areas of blanks. The reasons are explained below.

In Sec. 3, we mentioned that loss surfaces often demonstrate large regions of blanks. The reason is that the exponential and log operation involved in the cross entropy calculation. In this section, we give a simple case to show the ineffectiveness and demonstrate how blank regions produce. Consider a 10-class neural network and one input image of label 0. Suppose we have ten different logit output as $[0, 1, \cdots, 1], [1, 1, \cdots, 1] \cdots [9, 1, \cdots, 1]$ with confidence score ranged from -1 to 8. The corresponding cross entropy loss for ten different predictions are $[3.23, 2.30, 1.46, 0.79, 0.37, 0.15, 0.05, 0.02, 0.01, 0.01]$. We could see that in low confidence cases, cross entropy loss demonstrate informative trends with the increase of confidence. But when neural network prediction confidence reaches or above 5, the loss hardly changes, which causes certain large blank regions when visualizing the loss surfaces, which is consistent with Fig. 11.

## 8.2 DECISION SURFACE VISUALIZATION WITH MORE INPUT POINTS

### 8.2.1 COMPARISONS OF LOSS SURFACE AND DECISION SURFACE WITH MNIST INPUT IMAGES

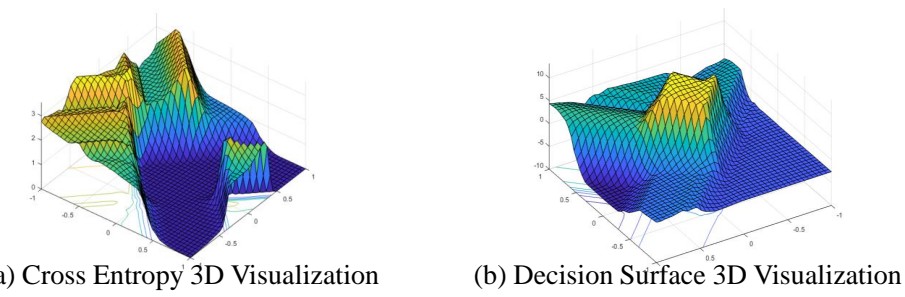

    (a) Cross Entropy 3D Visualization       (b) Decision Surface 3D Visualization

Figure 12: Comparison of cross entropy loss surface and decision surface of state-of-the-art robust model trained by MinMax robust training. We can see that cross entropy based loss surfaces demonstrate the opposite geometry with decision surface, but decision surfaces are more stable and clear in both high confidence areas and low confidence areas.

### 8.2.2 COMPARISONS OF ROBUST AND NATURAL MODELS WITH CIFAR INPUT IMAGES

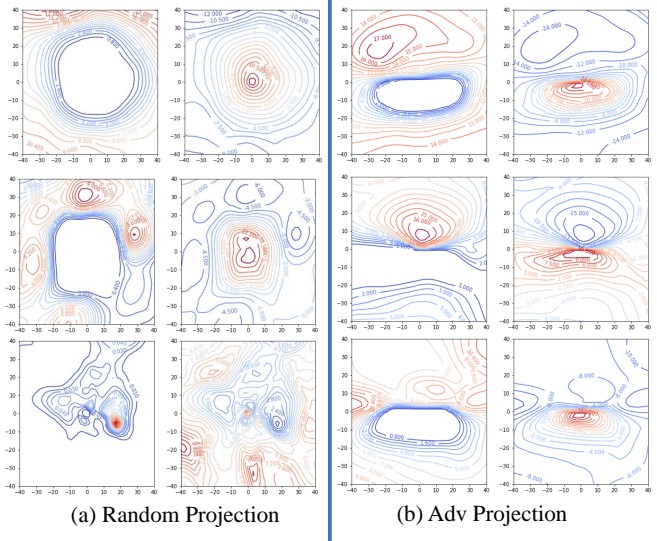

(a) Random Projection      (b) Adv Projection

Figure 13: Comparison of Natural Projection and Adversarial Projection on Natural Models. We could find that even with random projections, the loss and decision surfaces could demonstrate highly non-smooth patterns. This neighborhood under-fitting issue is the cause of adversarial examples.

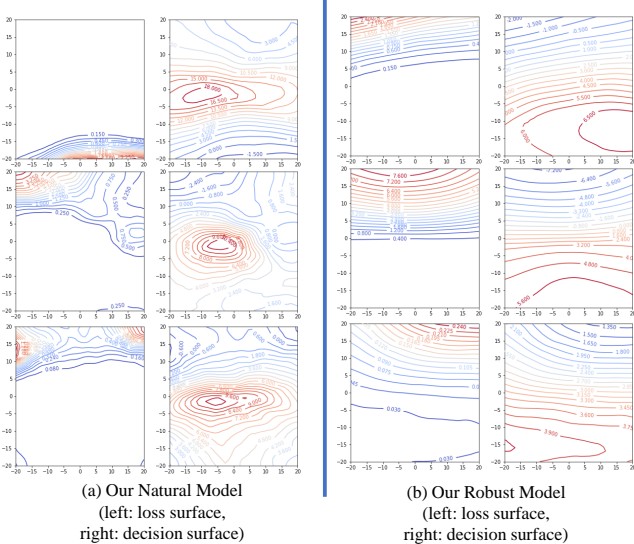

(a) Our Natural Model
(left: loss surface,
right: decision surface)

(b) Our Robust Model
(left: loss surface,
right: decision surface)

Figure 14: Comparison of Adversarial Projections on Natural and our Robust Models. We could find that the gradient regularization indeed smooths the network decision surfaces, bringing lower slopes. Therefore, adversarial attacks need to add step size to change the neural network decision.

## 8.3 EXPERIMENT SETTINGS

In the evaluation on MNIST dataset, a four-layer neural network model with two convolutional layers and two fully connected layers is adopted. After natural training, the baseline model achieves 99.17% accuracy. And for CIFAR10, we use a regular ConvNet with five convolutional layers and one global average pooling layer. For iterative methods BIM and C&W attack, we use 10 iterations and step size = 0.1 on MNIST and 1 on CIFAR. In adversarial training method, we use C&W attack to generate adversarial examples: For MNIST, we use 10 iterations, step size = 0.1, $\ell_\infty = 0.3$ on pixel range $0 \sim 1$. For CIFAR10, we use 10 iterations, step size = 1, $\ell_\infty = 8$ on pixel range $0 \sim 255$. The gradient regularization coefficient c (in Eq. 12) is set to 500 for gradient regularization.

## 8.4 INTERPRETABILITY OF JACOBIAN OF OUR ROBUST MODELS

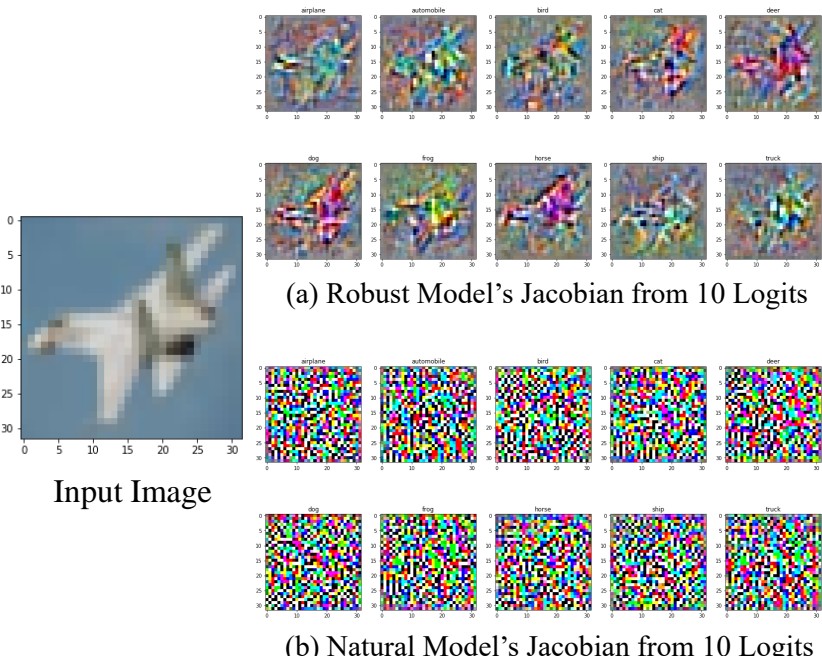

Input Image

(a) Robust Model's Jacobian from 10 Logits

(b) Natural Model's Jacobian from 10 Logits

Figure 15: Comparison of robust model's and natural model's Jacobian Visualization. (Both normalized to $0 \sim 255$ for visualization.) As we declared before, the property that Jacobian concentrates on the main pattern of images could enable the neural network better resistance with adversarial noises.

## 8.5 MORE PARAMETER SPACE EXAMPLES OF NATURAL VS. ROBUST MODELS

In this section, we provide more parameter space loss surface examples of natural and robust models (Madry et al. (2018)). Specifically, there are four possible combinations of parameter space settings: 1. natural model surface on natural input; 2. natural model surface on adversarial input; 3. robust model surface on natural input; 4. robust model surface on adversarial input. Here we show more examples of parameter space in the above four situations. Here natural input is one random batch (10 images from CIFAR10 test set), and adversarial input are FGSM (Kurakin et al. (2016)), Least-Likely attack (Kurakin et al. (2016)), and C&W (Carlini & Wagner (2017)) adversarial examples with eps=3 (Least-Likely and C&W use 10 iterations with step=1). All parameter space cross-entropy loss visualization results are shown in Fig. 16.

As expected, in parameter space, natural model's loss surface on adversarial inputs has a larger base height than the robust model, i.e. the average loss values are much higher than robust models. But the gap is only obvious on weak attacks, like FGSM, as shown in Fig. 16 (b). When we use strong attacks like C&W in Fig. 16 (d), the loss surfaces of the natural model and robust model become similar again: Both models' surfaces demonstrate high cross-entropy loss.

Therefore, we can use the loss surface in weight space to show their robustness difference if they are both plotted with weak adversarial inputs. But when we are facing stronger iterative attacks, the loss surface in weight space can no longer show any difference, thus cannot indicate the model robustness. By contrast, our input space loss surfaces can explicitly show the model robustness difference with no such restrictions, and the robustness difference can also be more obviously demonstrated, as shown in main paper Fig.3. And thus we believe these are the advantages of using input space loss surface to indicate the model robustness.

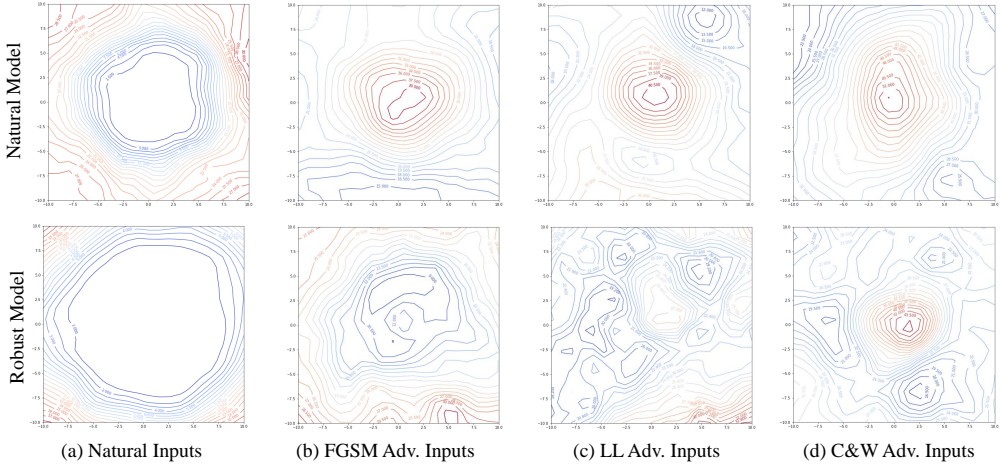

|  | (a) Natural Inputs | (b) FGSM Adv. Inputs | (c) LL Adv. Inputs | (d) C&W Adv. Inputs |

Figure 16: More examples of natural model's and robust model's loss surfaces in parameter space. As expected, robust model's loss surface in parameter space is more stable under weak attacks compared to the natural model, as shown in (b) and (c). (In case of misunderstanding, we note that the blue region is low loss region, and the red region is high loss region. The "more stable" means loss increment is less in robust model's loss surface, when comparing the second row to the first row.) But under stronger attacks, like C&W attack in (d), robust model's and natural model's loss surfaces in parameter space become similar again. Therefore, parameter space loss surfaces may not be suitable to indicate robustness well under such strong attack situations.

## 8.6 ADDITIONAL MNIST AND CIFAR10 EXPERIMENTAL RESULTS

Due to space limit, we add the left MNIST and CIFAR10 experimental results in Table. 3 and Table. 4 here.

Table 3: Test Accuracy of adversarial examples on MNIST dataset (%)

| Models | Natural | FGSM | | | BIM | | | C&W | | |
|---|---|---|---|---|---|---|---|---|---|---|
| | | 0.1 | 0.2 | 0.3 | 0.1 | 0.2 | 0.3 | 0.1 | 0.2 | 0.3 |
| Ours+AdvTrain | 95.9 | 87.6 | 72.2 | 44.1 | 89.2 | 67.2 | 28.4 | 89.6 | 73.2 | 39.5 |
| MinMax | 98.3 | 97.3 | 96.3 | 95.2 | 97.2 | 94.3 | 92.8 | 97.6 | 96.4 | 94.5 |

Table 4: Original MinMax Model's Test Accuracy of adversarial examples on CIFAR dataset (%)

| Model | Natural | FGSM | | | | BIM | | | | C&W | | | |
|---|---|---|---|---|---|---|---|---|---|---|---|---|---|
| | | 3 | 6 | 8 | 9 | 3 | 6 | 8 | 9 | 3 | 6 | 8 | 9 |
| MinMax | 87.3 | 75.3 | 63.2 | 56.1 | 53.4 | 74.2 | 59.3 | 48.7 | 46.2 | 74.2 | 59.2 | 49.8 | 46.1 |

