# OpenReview forum: "Interpreting Adversarial Robustness: A View from Decision Surface in Input Space"
_ICLR.cc/2019/Conference_

### Official Review · AnonReviewer2 · 2018-11-05
**An interesting visualization paper, but not always so convincing**

**Rating:** 5
**Confidence:** 5

**Review:**

This paper uses visualization methods to study how adversarial training methods impact the decision surface of neural networks.  The authors also propose a gradient-based regularizer to improve robustness during training.

Some things I liked about this paper:
The authors are the first to visualize the "decision boundary loss".  I also find this to be a better and more thorough study of loss functions than I have seen in other papers.  The quality of the visualizations is notably higher than I've seen elsewhere on this subject.

I have a few criticisms of this paper that I list below:
1)  I'm not convinced that the decision surface is more informative than the loss surface.  There is indeed a big "hole" in the middle of the plots in Figure 4, but it seems like that is only because the first contour is drawn at too high a level to see what is going on below.  More contours are needed to see what is going on in that central region.
2) The proposed regularizer is very similar to the method of Ross & Doshi.  It would be good if this similarity was addressed more directly in the paper.  It feels like it's been brushed under the rug.
3) In the MNIST results in Table 1:  These results are much less extensive than the results for CIFAR.  It would especially be nice to see the MinMax results since those of commonly considered to be the state of the art. The fact that they are omitted makes it feel like something is being hidden from the reader.
4) The results of the proposed regularization method aren't super strong.  For CIFAR, the proposed method combined with adversarial training beats MinMax only for small perturbations of size 3, and does worse for larger perturbations.  The original MinMax model is optimized for a perturbation of size 8.  I wonder if a MinMax result with smaller epsilon would be dominant in the regime of small perturbations.

---

> ### Author Response · Authors · 2018-11-06
> **Answers to reviewer's concerns**
>
> 5)	And we still need to claim two points about why our method is still meaningful here:
> a)	The training cost of MinMax makes it hard to scale. MinMax is commonly known that this method cannot generalize to large-scale datasets [2], e.g. ImageNet, since every training step in MinMax needs to generate PGD adversarial examples through 10-30 backpropagations. This makes training large-scale MinMax robust models impractical. But our method has better scaling ability than MinMax, the time consumption by double-backpropagation per training step is about 2.1 times than normal training, which is thus 5-15 times less than MinMax.
> b)	Meanwhile, on CIFAR10, the gap between MinMax and our method is not that large. Especially, the robustness gap under eps=3 attacks (FGSM, BIM, C&W) is negligible as shown in Table.2.  And about the robustness degradation under larger step attack, our reason analysis is stated in Sec 5.2, that because Taylor Approximation performs well in a small neighborhood but has limitations against larger step attack, which is a limitation of our method which we also talked in the paper.
>
> 6)	Lastly, as our paper named "Interpreting Adversarial Robustness: A View from Decision Surface in Input Space", we sincerely hope that reviewer could also take our paper’s other contributions into consideration, like revealing the nature of adversarial examples and robustness are actually solving NN’s neighborhood underfitting issue, the shared mechanism of various adversarial attacks by decision loss surface visualization and interpretation, proof of the relationship between loss geometry and adversarial robustness by Jacobian and Hessian’s geometry properties, etc., and we believe our paper is a thorough analysis and interpretation work in current interpreting adversarial attack and robustness research.
>
> Again, we thank the reviewer for the detailed reviews!
>
> [1] Andrew Slavin Ross and Finale Doshi-Velez. Improving the adversarial robustness and interpretability of deep neural networks by regularizing their input gradients. In AAAI, 2018.
> [2] Kannan, Harini, Alexey Kurakin, and Ian Goodfellow. "Adversarial Logit Pairing." arXiv preprint arXiv:1803.06373 (2018).

---

> > ### Comment · AnonReviewer2 · 2018-12-05
> > **Points taken**
> >
> > I agree that adversarial training is expensive and the proposed method offers a cheaper alternative.  However I still think the related work needs to be discussed and compared to, given the strong similarities to the other Jacobian regularizers in the literature.
> >
> > I acknowledge the paper's various contributions.  To be clear, I think this paper hits on a number of interesting issues.  However I think some of the experiments feel incomplete or hard to interpret.  I also think the evaluation of the proposed method is incomplete.  I'd like to know how it compared to other Jacobian regularizers.  Does regularizing the decision loss Jacobian really beat the cross entropy loss?  I know there's a conceptual argument to be made in support of the decision loss, but I just don't feel like the experiments currently back that up.  For this reason the paper feels a bit incomplete to me.

---

> ### Author Response · Authors · 2018-11-06
> **Answers to reviewer's concerns**
>
> We have updated our submitted paper and added the MNIST and CIFAR experimental results in Appendix 8.6.
> Thanks a lot for the reviewer's suggestions for our experiments!
>
> -----------------------------------------------------------------------------------------------------------------------------------------------------------------
>
> We thank the reviewer for your interests in our visualization results!
>
> 1)	About your first concern on our statement, that decision loss is more informative, there are two reasons:
> a)	About the big hole, we give a simple example and two illustration figures in Appendix 8.1. The “less informative” blank region is caused by the non-linear operations (soft-max and entropy). Specifically, when a neural network has high confidence in correct logits, the cross-entropy could hardly describe the confidence information:
> For example, ten-class model with nine pairs of different logit outputs [1, 1, 1, …, 1, 1], [2, 1, 1, …, 1, 1], …. [9, 1, 1, …, 1, 1];
> The corresponding cross-entropy loss is [2.30, 1.46, 0.79, 0.37, 0.15, 0.05, 0.02, 0.01, 0.01];
> The corresponding confidence defined in Eq.2 is [0, 1, 2, 3, 4, 5, 6, 7, 8];
> With further increasing of logits confidence (model confidence could easily achieve over 20 in common NN models), the cross-entropy loss hardly changes anymore. But the confidence’s change is stable. That’s why the blank region appears in cross-entropy loss surface but not in decision surface. Therefore, we state the decision confidence surfaces could provide more geometry information in the “blank region” of cross-entropy loss surface.
> b)	Second, the decision boundary loss surface contains the explicit decision boundary (contour line L=0), across which the model’s prediction result will be flipped. By contrast, cross-entropy loss surface has no such explicit decision boundary. This is one very important property since this enables us to visualize and evaluate the attack strength needed to conduct a successful adversarial attack against the model.
> Therefore with these two points, we claim that decision surface is more informative than cross-entropy loss surface.
>
> 2)	About how to distinguish our work from Ross & Doshi [1], the same part is we use the same Loss_ce + Loss_grad idea but different regularizer design in Loss_grad. We choose to penalize decision boundary loss’s (in Eq.2) gradients while Ross & Doshi use common cross-entropy loss’s gradients. The benefit of doing so is related to the problem we mentioned in above part 1(a). Because of the cross-entropy loss involves highly non-linear soft-max and entropy operation compared to the decision loss, the changing of cross-entropy loss is negligible in high confidence cases (as we mentioned before) but decision loss has no such drawbacks. The non-linear operations will also cause the gradient of cross-entropy loss is relatively small, which brings constraints on the gradient penalty effect. As the comparison experiments in Table1 and 2 showed, our decision loss gradient regularizer outperforms cross-entropy loss regularization a lot on both MNIST and CIFAR10.
>
> 3)	About the 3rd and 4th concerns, the reason we omit the MinMax model on MNIST is purely space consideration. Here we comment with the missing part our Ours+Adv Training, and MinMax’s experimental results. We will update our version added with these MNIST experimental results.
>
> Attack | Natural   ----FGSM----              ----BIM----              ----C&W----
> Epsilon	|    0      |0.1|0.2|0.3|           0.1|0.2|0.3|           0.1|0.2|0.3
> Ours+Adv|  95.9 | 87.6|72.2|44.1|   89.2|67.2|28.4|   89.6|73.2|39.5
> MinMax	|  98.4   |97.3|96.3|95.2|   97.2|94.3|92.8|    97.6|96.4|94.5
>
> The MinMax model is absolutely the most robust model, which is the reason why we choose its model to analyze the decision surface geometry. In Fig.6, the (eps < 0.3) region of MinMax model’s decision surface is nearly flat with no downhills under both random noises and adversarial attacks, which makes it near immune to adversarial attacks.
>
> 4)	And about the final concern that on CIFAR10, will original MinMax released model be dominant in small perturbation attacks, below we show the original released model’s (trained with eps=8) performance under the attacks:
>
> Attack	Natural	        ----FGSM----                       ----BIM-----	                           ---C&W---
> Epsilon	0	               | 3| 6|8|9|	                    |3|6|8|9|	                          |3|6|8|9|
> Acc	       87.3	          |75.3|63.2|56.1|53.4|     |74.2|59.3|48.7|46.2|     |74.2|	59.2|49.8|46.1|
>
> The released original model did not show dominant accuracies on small perturbations above our methods on CIFAR10. The robustness gap is within 10% robustness excluding the model baseline accuracy difference (baseline accuracy difference is about 4% compared to Ours+AdvTrain).

---

> > ### Comment · AnonReviewer2 · 2018-12-05
> > **I'm unclear about the response to comment 1 & 4**
> >
> > 1)  I see your point about the decision surface. You make an interesting argument that changes in the loss can be quite small when confidence increases.  However, it could be that small changes in the confidence are meaningful even though they're small.  I *still* want to see what in the blank holes in Figure 4.  I understand the the loss is small in this region, however the only reason there's a blank region is because of how you (arbitrarily) chose to draw the contours at too high a level to see what's going on.
> >    You make an argument here that the decision surface is more informative because the gradient direction contains more information than the loss gradient.  However, I can't see the loss gradient in your figure because the contours start at too high a value, so I don't see how this figure backs up this hypothesis.
> >
> > 4) My concern was that the original MinMax model is optimized for epsilon=8.  The method "outs+advtrain" in Table 2 out-performs MinMax for small epsilon.  However the table might turn if you compared against a MinMax optimized for smaller epsilon (in this case MinMax might beat the proposed method for epsilon=3).  Could you please clarify how the results  you show in your rebuttal differ from the paper?  You seem to claim this model is *also* trained with epsilon-8, so what did you change?

---

> > > ### Author Response · Authors · 2018-12-05
> > > **Reply to Reviewer 2**
> > >
> > > Thanks a lot for your review!
> > >
> > > 1. About the first concern, we understand your concerns that the cross-entropy loss surface hole might be caused by drawing the plots too high to see that the gradients' direction. We will try to lower the magnitude of the loss surface and visualize it again.
> > > But still, our point that decision loss is more informative here is not only about the gradient's direction,  but also the changing speed of the loss. And the example we give in the appendix shows, the cross-entropy loss will hardly change in high-confidence areas, which causes that it is not suitable for visualization in 2D and 3D cases, e.g. in Fig.12 (a).
> > >
> > > 2.  In the original paper, our CIFAR10 results are obtained on a ConvNet (which we have given in the first version in Appendix Sec. 8.3). And the further provided results in rebuttal (Appendix in Table.4 ) is the original MinMax's ResNet model's test results, as required by the reviewer.  That's why they are different.
> > >
> > > Hope this clarifies your concerns. Thanks a lot for the reviews!

---

### Official Review · AnonReviewer1 · 2018-11-06
**good visualization for adversarial robustness analysis, unclear loss surface in weight space with adversarial data**

**Rating:** 6
**Confidence:** 4

**Review:**

The authors demonstrated that the loss surface visualized in parameter space does not reflect its robustness to adversarial examples. By analyzing the geometric properties of the loss surface in both parameter space and input space, they find input space is more appropriate in evaluating the generalization and adversarial robustness of a neural network. Therefore, they extend the loss surface to decision surface. They further visualized the adversarial attack trajectory on decision surfaces in input space, and formalized the adversarial robustness indicator. Finally, a robust training method guided by the indicator is proposed to smooth the decision surface.

This paper is interesting and well organized. The idea of plotting loss surface in input space seems to be a natural extension of the loss surface w.r.t to weight change. The loss surface in input space measures the network’s robustness to the perturbation of inputs, which naturally shows the influence of adversarial examples and is suitable for studying the robustness to adversarial examples.

Note that loss surface in parameter space measures the network’s robustness to the perturbation of the weights with given inputs, which implicitly assumes the data distribution is not significantly changed so that the loss surface may have similar geometry on the unseen data.

The claim of “these significant distinctions indicate the ineffectiveness of generalization estimation from the loss surfaces in parameter space” are not well supported as the comparison between Figure 2(a) and Figure 3 seems to be unfair and misleading. Fig 2 are plotted based on input data without any adversarial examples. So it is expected to see that Fig 2(a) and Fig 2(b) have similar contours. However, the loss surface in weight space may still be able to show their difference if the they are both plotted with the adversarial inputs. I believe that models trained by Min-Max robust training will be more stable in comparison with the normally trained models. It would be great if the author provide such plots. I would expect the normal model to have a high and flat surface while the robust model shows reasonable loss with small changes in weight space.

How to choose \alpha and \beta for loss surface of input space for Fig 3 and Fig 4 (first row)?

How are \alpha and \beta normalized for loss surface visualization in weight space as in Eq 1?

---

> ### Author Response · Authors · 2018-11-06
> **Reply to "good visualization ..."**
>
> We have updated our submitted paper to address your concerns in Sec.2.2 and in Appendix 8.5.
> Thanks a lot for the reviewer's suggestions!
>
> --------------------------------------------------------------------------------------------------------------------------------------------------------------
>
> We thank the reviewers for liking the visualization idea!
>
> 1.	About your first concern that “The claim of ‘these significant distinctions indicate the ineffectiveness of generalization estimation from the loss surfaces in parameter space’ are not well supported”, please let us provide some clarification. We have provided the natural model and robust MinMax model’s contour maps on natural input and adversarial inputs and their Visualizations in the Appendix (the updated version) 8.5, Fig.16.
>
> As expected, in parameter space, natural model's loss surface on adversarial inputs has a larger base height than the robust model, i.e. the average loss values are higher than robust models. But such gap is only obvious on weak attacks, like FGSM. When we use stronger attacks like C\&W attack, the loss surface of the natural model and robust model become similar again: Both models' surfaces demonstrate high cross-entropy loss with no obvious distinction.
>
> Therefore, as mentioned in the review, we can indeed use the loss surface in weight space to show their robustness difference if they are both plotted with weak adversarial inputs. But when we are facing stronger iterative attacks, the loss surface in weight space can no longer show any difference, thus cannot indicate the model robustness.
>
> By contrast, our input space loss surfaces can explicitly show the model robustness difference with no such restrictions, and the robustness difference can also be more obviously demonstrated, as shown in main paper Fig.3. Therefore, we think this is the advantage of using input space loss surface to indicate the model robustness. We will absolutely update the statement in the main paper.
>
> 2.	In Fig.3, both x-axes (alpha) in (a) and (b) are chosen as random directions with normalization, and both y-axes (beta) are chosen as FGSM attack direction [1].
> In Fig.4, all x-axes (alpha) in (a)-(d) are chosen as random directions with normalization, and y-axes are as following (formulas in Eq.3): (a) random direction, (b) FGSM attack direction [1], (c) Least-likely class attack direction [1], (d) C&W attack direction [2], all with normalization.
>
> 3.	The gradients and random noises are normalized as in Fast gradient sign method [1], which is the pixel-wise sign, i.e. beta = sign(beta).
>
> We thank the reviewer for the constructive comments on the first point, and we will absolutely update the statement to be more accurate.
>
> [1] Adversarial examples in the physical world, Alexey Kurakin et al, 2016.
> [2] Towards evaluating the robustness of neural networks. Carlini, Nicholas, and David Wagner. IEEE (SP), 2017.

---

> > ### Comment · AnonReviewer1 · 2018-11-21
> > **Figure 16 shows that loss surface in weights space still works for adversarial examples in certain cases.**
> >
> > Thanks for the clarification. Figure 16 is a good example showing that loss surface in weights space still works for some adversarial attacks. Actually for the strong attacks show in Figure 16(d), the loss surface of robust model seems seems still better than the natural model as the red contour area is much smaller.
> >
> > Figure 16 is actually more informative than Figure 2. I would suggest the authors to make it clear in the main text about the results provided by Figure 16, i.e., the loss surface in weight space does not always fail for showing model's robustness to adversarial attacks. More in-depth analysis about the reason for this result could be better.
> >
> > I was asking how are x-axes(\alpha) and y-axes (\beta) normalized for loss surface visualization in weight space for Fig. 2, Fig.11 and Fig 16? As indicated in [1], different normalization method used could show different width of loss contours. I was wondering whether the blank space is caused by different normalization method.
> >
> > [1] Visualizing the loss landscape of neural nets, Li et al, NIPS, 2018

---

> > > ### Author Response · Authors · 2018-11-21
> > > **About 'Normalization and Blank space'**
> > >
> > > Thanks for the reviewer's comments! Here are our clarifications:
> > >
> > > 1. The normalization in weight space is also done by sign(*) operation.  But the step size here is different (previously in input image space, step_size=1 pixel), here step_size is set to 0.01. The normalization is done by  following formula:   \alpha = 0.01 * sign(\alpha). (same as \beta)
> > >
> > > 2. About the blank area question, I think there might be some misunderstanding here. Fig.11 shows loss surfaces and decision surfaces both in input space instead of parameter space.  And Appendix-1 is trying to explain the reason of why the blank area of loss surface in input space produces, and thus why the decision surface is better than loss surface.
> > >
> > > 3. About the different normalization methods, [1] proposed that different normalization could influence the width of the loss surfaces in weight space. But using normalization method in 1. with a proper step size, the loss surface is already capable to show very different features in Fig.16 (a)-(d). Therefore, we didn't use [1]'s proposed method.
> > >
> > > Thank you very much for your comments. We will make Fig.11 clearer by adding "loss surfaces and decision surfaces (both in input space)"

---

### Official Review · AnonReviewer3 · 2018-11-12
**Review of "Interpreting Adversarial Robustness: A View from Decision Surface in Input Space"**

**Rating:** 3
**Confidence:** 5

**Review:**

This paper argues that analyzing loss surfaces in parameter space for the purposes of evaluating adversarial robustness and generalization is ineffective, while measuring input loss surfaces is more accurate. By converting loss surfaces to decision surfaces (which denote the difference between the max and second highest logit), the authors show that all adversarial attack methods appear similar wrt the decision surface. This result is then related to the statistics of the input Jacobian and Hessian, which are shown to differ across adversarially sensitive and robust models. Finally, a regularization method based on regularizing the input Jacobian is proposed and evaluated. All of these results are shown through experiments on MNIST and CIFAR-10.

In general, the paper is clear, though there are a number of typos. With respect to novelty, some of the experiments are novel, but others, including the improved training method, have been explored before (see specific comments for references). Finally, regarding significance, many of the insights provided this paper are true by definition, and are therefore unlikely to have a significant impact.

While I strongly believe that rigorous empirical studies of neural networks are essential, this paper is lacking in several key areas, including framing, experimental insights, and relation to prior work, and is therefore difficult to recommend. Please see the comments below for more detail.

Major comments:

1) In the beginning of the paper, adversarial robustness and generalization are equated. However, adversarial robustness and generalization are not necessarily equivalent, and in fact, several papers have provided evidence against this notion, showing that adversarial inputs are likely to be present even for very good models [5, 6] and that adversarially sensitive models can often generalize quite well [2]. Moreover, all the experiments within the paper only address adversarial robustness rather than generalization to unperturbed samples.

2) One of the main results of this paper is that the loss surface wrt input space is more sensitive to adversarial perturbations than the loss surface wrt parameter space. Because adversarial inputs are defined in input space, by definition, the loss surface wrt to the input must be sensitive to adversarial examples. This result therefore appears true by definition. Moreover, [3] related the input Jacobian to generalization, finding a similar result, but is not discussed or cited.

3) The main result of Section 3 is that all adversarial attacks “utilize the decision surface geometry properties to cross the decision boundary within least distance.” While to my knowledge the decision surface visualization is novel and might have important uses, this statement is again true by definition, given that adversarial attack methods try to find the smallest perturbation which changes the network decision. As a result, all methods must find directions which are short paths in the decision surface. It is therefore unclear what additional insight this analysis presents.

4) How does measuring the loss landscape as an indicator for adversarial robustness differ from simply trying to find adversarial examples as is common? If anything, it seems it should be more computationally expensive as points are sampled in a grid search vs optimized for.

5) The proposed regularizer for adversarial robustness, based on regularizing the input Jacobian, is very similar to what was proposed in [1], yet [1] is not discussed or cited.

Minor comments:

1) The paper’s first sentence states that “It is commonly believed that a neural network’s generalization is correlated to ...the flatness of the local minima in parameter space.” However, [4] showed several years ago that the local minima flatness can be arbitrarily rescaled and has been fairly influential. While [4] is cited in the paper, it is only cited in the related work section as support for the statement that local minima flatness is related to generalization when this is precisely opposite the point this paper makes. [4] should be discussed in more detail, both in the introduction and the related work section.

2) The paper is quite lengthy, going right up against the hard 10 page limit. While this may be acceptable for papers with large figures or which require the extra space, this paper does not currently meet that threshold.

3) Throughout the figures, axes should be labeled.

4) In section 2.2, it is stated that both networks achieve optimal accuracy of ~90% on CIFAR-10. This is not optimal accuracy and hasn’t been for several years [7].

5) Why is equation 2 calculated with respect to the logit layer vs the normalized softmax layer? Using the unnormalized logits may introduce noise due to scaling.

6) In Figure 8, the scales of the Hessian are extremely different. Does this impact the measurement of sparseness?


Typos:

1) Introduction, second paragraph: “For example, ResNet model usually converges to…” should be “For example, ResNet models usually converge to…”

2) Introduction, second paragraph: “...defected by the adversarial noises...” should be “...defected by adversarial noise…”

3) Introduction, third paragraph: “...introduced by adversarial noises...” should be “...introduced by adversarial noise…”

4) Section 3.1, first paragraph: “cross entropy based loss surface is…” should be “cross entropy based loss surfaces is…”

[1] Jakubovitz, Daniel, and Raja Giryes. "Improving DNN Robustness to Adversarial Attacks using Jacobian Regularization." arXiv preprint arXiv:1803.08680 (2018). ECCV 2018.
[2] Zahavy, Tom, et al. "Ensemble Robustness and Generalization of Stochastic Deep Learning Algorithms." arXiv preprint arXiv:1602.02389 (2016). ICLR Workshop 2018
[3] Novak, Roman, et al. "Sensitivity and generalization in neural networks: an empirical study." arXiv preprint arXiv:1802.08760 (2018). ICLR 2018.
[4] Dinh, Laurent, et al. "Sharp Minima Can Generalize For Deep Nets." International Conference on Machine Learning. 2017.
[5] Fawzi, Alhussein, Hamza Fawzi, and Omar Fawzi. "Adversarial vulnerability for any classifier." arXiv preprint arXiv:1802.08686 (2018). NIPS 2018.
[6] Gilmer, Justin, et al. "Adversarial spheres." arXiv preprint arXiv:1801.02774 (2018). ICLR Workshop 2018.
[7] http://rodrigob.github.io/are_we_there_yet/build/classification_datasets_results.html

---

> ### Author Response · Authors · 2018-12-17
> **Reply to reviewer3**
>
> Thanks for the reviewer’s precious comments! We do believe that there is a lot to be improved in this paper!
> 1)	About the first concern of the reviewer, the reason of “equated generalization” is that previously upon the finish of this paper, there is no clear definition of the generalization in the adversarial settings. And our statement is actually saying that “adversarial robust generalization”  doesn’t equal to “standard generalization”, as in the NeurIPS’18 tutorial slides, https://media.neurips.cc/Conferences/NIPS2018/Slides/adversarial_ml_slides_parts_1_4.pdf, page 29-30. We can now give the formal definition maybe in the future paper, thanks for the advice.
> 2)	For the second concern, by briefly comparing weight/input space visualization, the main conclusion we want to draw is that, past generalization analysis cannot be well adopted in the adversarial settings. I think this should make more sense.
> 3)	About the sec.3, we don’t agree with the reviewer that the visualization insights are trivial because the visualization results are one of the key intuition, which is “The geometry slopes (gradients) matter a lot in the model’s robustness” and this is the motivation of why regulating jacobian could improve the robustness in the whole paper.
> 4)	About the robustness indicator, we have shown a case study in the Sec4.3 ROBUSTNESS INDICATOR EVALUATION. We could compare two model’s Jacobian & Hessian to distinguish two model’s robustness easily, as shown in Fig.8. This is not done by grid search, but by backpropagation to compute Jacobian and Hessian, therefore not computationally expensive.
> 5)	About the Jacobian Regulation novelty, we do agree that the robust training part is not a significant contribution, as also mentioned by reviewer-2. Our reply about the difference is here: https://openreview.net/forum?id=rylV6i09tX&noteId=HJgm8ciA3X under the reviewer-2’s comments.
> 6)	About the related work and other minor comments, thanks again for the precious comments!

---

### Meta-Review · Area_Chair1 · 2018-12-13

**Confidence:** 5
**Recommendation:** Reject

**Metareview:**

This paper studies the relationship between flatness in parameter space and generalization. They show through visualization experiments on MNIST and CIFAR-10 that there is no obvious relationship between the two. However, the reviewers found the motivation for the visualization approach unconvincing and further found significant overlap between the proposed method and that of Ross & Doshi. Thus the paper should improve its framing, experimental insights and relation to prior work before being ready for publication.